# Human Body-Fluid-Assisted Fracture of Zinc Alloys as Biodegradable Temporary Implants: Challenges, Research Needs and Way Forward

**DOI:** 10.3390/ma16144984

**Published:** 2023-07-13

**Authors:** R. K. Singh Raman, Cuie Wen, Jörg F. Löffler

**Affiliations:** 1Department of Mechanical & Aerospace Engineering, Monash University, Clayton, VIC 3800, Australia; 2Department of Chemical & Biological Engineering, Monash University, Clayton, VIC 3800, Australia; 3School of Engineering, RMIT University, Melbourne VIC 3001, Australia; cuie.wen@rmit.edu.au; 4Laboratory of Metal Physics and Technology, Department of Materials, ETH Zurich, 8093 Zurich, Switzerland; joerg.loeffler@mat.ethz.ch

**Keywords:** zinc alloys, biodegradable implants, human body fluid, stress corrosion cracking, corrosion fatigue, magnesium alloys, iron alloys

## Abstract

Alloys of magnesium, zinc or iron that do not contain toxic elements are attractive as construction material for biodegradable implants, i.e., the type of implants that harmlessly dissolve away within the human body after they have completed their intended task. The synergistic influence of mechanical stress and corrosive human body fluid can cause sudden and catastrophic fracture of bioimplants due to phenomena such as stress corrosion cracking (SCC) and corrosion fatigue (CF). To date, SCC and CF of implants based on Zn have scarcely been investigated. This article is an overview of the challenges, research needs and way forward in understanding human body-fluid-assisted fractures (i.e., SCC and CF) of Zn alloys in human body fluid.

## 1. Introduction

Avoiding an orthopedic surgical procedure is an extremely attractive proposition in addressing health concerns of ageing populations, which are among the greatest challenges of current times. A surgical procedure can be circumvented when temporary implants (such as pins, wires, screws, plates, etc.) are constructed from alloys that are not only non-toxic and biocompatible but can also harmlessly dissolve within the human body after accomplishing their temporary healing function [1,2,3,4,5]. This phenomenon was demonstrated through in vivo tests [5], where pins of a magnesium (Mg) alloy ZX50 implanted into femoral bone tissues of Sprague–Dawley rats were observed after 4, 12, 24 and 36 weeks. The pins were found to have completely dissolved and the fractured bone healed after 36 weeks, as shown in Figure 1 [5]. Thus, constructing implants from such alloys can avoid the burdensome procedure of a second surgery, which is often undertaken to remove a temporary implant when it is constructed out of commonly used traditional materials, e.g., titanium alloys or stainless steels.

Among the structural metals, Mg, Zn and Fe meet the criteria of non-toxicity and biodegradability, which are essential requirements for temporary biodegradable implants. The objective of this article is to present an overview of the comparative merit of zinc (Zn) alloys vis-à-vis magnesium (Mg) and iron (Fe) alloys for non-toxic, biodegradable, temporary implant applications, followed by a thorough discussion on the challenges, research needs and way forward in understanding the human body-fluid-assisted fracture (i.e., stress corrosion cracking and corrosion fatigue) of Zn alloys in human body fluid.

## 2. Biodegradable Temporary Implants: Critical Requirements and Challenges

Constructing temporary orthopedic implants from alloys that can harmlessly degrade away in the human body after completing their function has attracted rapidly increasing research and clinical interest. The most significant aspect of such application is to allow the alloy (containing only non-toxic elements) to slowly degrade/dissolve within the human body thus avoiding the second surgery that is usually required to remove the implant.

The most critical requirements of biodegradable alloys as temporary implants are the non-toxicity of the base metal and its alloying constituents to human physiology, and their ability to degrade away eventually, completely and harmlessly in the human body. The essential requisites of alloys for such applications include:(a)They should not suffer premature failure due to excessive corrosion [1,2,3,6];(b)Their elastic modulus (stiffness) should be close to that of human bone in order to avoid “stress shielding” during the healing process [7,8]. Stress shielding occurs when some of the stress is taken by the implant and thus shields the surrounding bone from developing the ability to sustain the stress. This may cause bone atrophy, implant loosening, and eventually, premature implant failure;(c)They should possess sufficient mechanical integrity during the entire span of their use as implant [6];(d)They must be resistant to sudden brittle fractures that may occur due to the synergy of mechanical stresses and the corrosive environment of human body fluid, i.e., stress corrosion cracking (SCC) and corrosion fatigue (CF) [4,8,9,10].

It is essential to ensure the mechanical integrity of the implant device, and that it possesses the desired level of resistance to corrosion and corrosion-assisted fracture for the duration of its temporary use. Therefore, selection of an alloy for temporary implant applications is governed by the constraints of the conflicting requirements of non-toxicity, biocompatibility, adequate mechanical properties and corrosion resistance in human body fluid. Such constraints constitute a narrow window of suitable metals and alloys, and alloys based on Mg, Zn and Fe are the most suitable metallic materials that meet these constraints. However, even for such alloys, some of the requirements may be challenging. For example, precipitation hardening that is commonly employed for strengthening is often deleterious for corrosion and corrosion-assisted premature fracture of Mg alloys, because the strengthening precipitates that are generally highly cathodic to the alloy matrix may cause localized pitting corrosion (as seen in Figure 2) [11]. Pits are well-known initiation sites for environment-assisted fracture, such as SCC and CF of such alloys [9,10]. Human body-fluid-assisted fracture (such as SCC and CF) is among one of the critical concerns [12,13,14] in the use of traditional implants. While this concern has often been resoundingly confirmed [9,10,15], it is a vastly underexplored research area in the context of Zn and Mg alloys for biodegradable implants.

## 3. Zinc Alloys Vis-à-Vis Other Alloys for Biodegradable Implants

Mg-, Fe- and Zn-based alloys often meet some or sometimes all the required criteria for their use as non-toxic, biodegradable implants, as will be discussed in the following section.

### 3.1. Mg Alloys

Mg possesses the most suitable properties: (a) its low density (*ρ* = 1.74–2.0 g cm^−3^) and elastic modulus (*E* = 41–45 GPa) are respectively similar to those of human bones (*ρ* = 1.8–2.1 g cm^−3^; *E* = 3–20 GPa) [3,6]; (b) Mg is not only biocompatible/non-toxic but is also essential to the human metabolism as a cofactor for many enzymes [8], and the degradation product of Mg^2+^ ions stimulates tissue growth and healing [3]; (c) any excess of Mg^2+^ ions is harmlessly excreted via urine [3]; and (d) Mg alloys may be considerably less expensive than traditional implant alloys (for example, Ti and Co alloys). A general drawback of Mg alloys is that they may corrode too rapidly (faster than the time required for bone healing), as well as generate excessive hydrogen bubbles upon rapid degradation.

### 3.2. Fe Alloys

Fe is non-toxic, biocompatible and biodegradable in the human body. However, Fe alloys possess a much higher modulus (>>100 GPa) than human bone, creating a risk of “stress shielding”, which leads to weakening of the bone. Another problem with Fe alloys in this specific application is the slow degradation rate.

### 3.3. Zn Alloys

Zn corrodes much faster than Fe in the human body, and thereby offers the opportunity to circumvent the issue of the sluggish degradation rate of Fe in temporary implant applications. Both in vitro and in vivo tests suggest that Zn corrodes at considerably slower rates (0.014–0.03 mm y^−1^) [1] than Mg (>0.2 mm y^−1^) [16], which may circumvent the concern of prohibitively high corrosion rates of Mg alloy implants, especially those Mg implants that are not based on ultrahigh-purity Mg [17], and thus reveal extended localized corrosion. It is critically important that, unlike Mg corrosion where hydrogen generation is the predominant cathodic reaction, the cathodic reaction for Zn corrosion is oxygen reduction [18], which eliminates the risk of hydrogen bubble generation. However, the elastic modulus of Zn (*E* = 80–120 GPa) is not as close to that of bone (*E* = 3–20 GPa) as that of Mg (*E* = 41–45 GPa) [1], thus enhancing the concern of stress shielding.

Zn is essential for human physiology. An adult human contains 2–3 g of zinc. Zn is a stimulus for the growth and mineralization of bone tissue, and inhibits bone resorption [19]. The human body requires ~15 mg/day [20] of Zn for essential human physiological functions, such as enzymatic catalysis and crucial neuronal functions [19], and as a cofactor in enzymes and regulatory proteins. Zn ions play a crucial role in regulating arterial blood pressure [19]; in fact, arterial hypertension is symptomized by a decrease in Zn ions in bones, lymphocytes and serum, and by their increase in kidney, heart, liver, spleen and suprarenal glands [20]. An optimized Zn content is also crucial for the immune system, and both higher and lower Zn contents can disturb it [21].

Li et al. [22] reviewed the limitations in the mechanical properties of Zn alloys, and their strength may be a concern. However, suitable alloying, such as with Li and Mg, can bring about the required improvement in strength, whereas alloying with Cu, Al, and Mn can improve ductility. Thermomechanical treatments (such as extrusion and rolling) for grain refinement are other means for improving strength. Zn alloys can readily undergo natural ageing, deteriorating the mechanical properties, and this is of concern for low-temperature applications such as bioimplant applications. Poor creep resistance is another concern for Zn alloys, but this relates only to elevated temperature applications (and not bioimplants). There is little reported on the fatigue properties of Zn alloys for bioimplant applications.

The article by Li et al. [22] provides a comprehensive review of Zn alloys for their potential as a biodegradable implant, particularly from the perspective of biomechanical compatibility. The principal alloys among several Zn-alloy families investigated for biodegradable implant applications are based on additions of Mg [23,24], Mn [25], Ca [26], Ag [27], and Fe [28], and combinations thereof [29]. However, most of these alloys contain second-phase precipitates, which can cause localized pitting corrosion that may facilitate SCC/CF fracture of the bioimplant, and the corresponding risk for the patient.

## 4. Human Body-Fluid-Assisted Degradation of Zinc Alloys as Biodegradable Temporary Implants

As discussed in the preceding section, Zn alloys meet considerable requisites as a construction material for biodegradable implants. There are a few reports on chemical degradation (corrosion) of Zn alloys in simulated body fluid (SBF). The corrosion and degradation behavior of Zn and its alloys can be determined via in vitro and in vivo studies.

The corrosive environment significantly affects the corrosion process and degradation rate of biodegradable Zn alloys, which have been proposed for applications in bone fixation plates and screws, vascular stents, and guided bone regeneration membranes [29]. The service environments in these applications contain serum, plasma, and interstitial fluid. In general, simulated fluids such as NaCl solution, Ringer’s solution, phosphate-buffered saline (PBS), simulated body fluids (SBFs), Hank’s balanced salt solution (HBSS), and cell culture medium (minimum essential medium and Dulbecco’s modified Eagle medium) are the most commonly used media for testing the corrosion behavior of biodegradable metal implants. Mei et al. [30] performed a detailed review on selecting a medium for corrosion testing of bioresorbable Mg alloys. They concluded that corrosion tests performed in NaCl solutions can provide more information on the intrinsic corrosion behavior of the material than tests in SBF or protein-containing solutions that are strongly affected by media components. Furthermore, synthetic pH buffers such as tris(hydroxymethyl) aminomethane buffer with HCl (Tris-HCl) and 4-(2-hydroxyethyl)-1-piperazineethanesulfonic acid (HEPES) should not be included in the corrosive media for testing Mg alloys. On the other hand, Liu et al. [31] compared the effects of the different buffer systems of Tris-HCl, HEPES, and NaHCO_3_/CO_2_ on the biodegradation behavior of pure Zn in NaCl solution and SBF, and suggested that the pH of corrosive media that plays an important role in the formation of corrosion products can be mediated using the three buffer systems. Their results indicate that Tris-HCl suppresses the formation of corrosion products and accelerates the corrosion process in both NaCl and SBF, whereas HEPES and NaHCO_3_/CO_2_ promote the formation of a passive film and slow down the corrosion rate in NaCl solution, but inhibit the deposition of corrosion products and accelerate the corrosion process in SBF. Furthermore, SBF buffered with Tris-HCl or NaHCO_3_/CO_2_ has been proposed as a suitable test medium to assess the in vitro biodegradation behavior of Zn in stent applications. Recently, Wang et al. [32] confirmed that the addition of synthetic pH buffers of Tris-HCl or HEPES stabilizes the local pH at the interface of pure Zn, Zn–0.8 Mg, and Zn–0.8Ca in HBSS (pH = 7.4) at 37 °C under hydrodynamic conditions, and Tris-HCl or HEPES-buffered HBSS is recommended as a suitable medium for in vitro evaluation of Zn-based biodegradable alloys.

The in vitro corrosion performance of Zn alloys can be evaluated using electrochemical tests including potentiodynamic polarization (PDP) and electrochemical impedance spectroscopy (EIS) in SBF, according to ASTM G59-97. The in vitro degradation rate of Zn alloys can be determined by immersion of the samples in SBF for a defined duration, followed by weight-loss measurements. The commonly used SBF can be either a Hank’s solution, a phosphate-buffered saline (PBS), a Ringer’s saline solution, or a cell culture medium mixed with PBS [29]. The corrosion byproducts of Zn in SBF contain its oxide (ZnO) and other compounds including ZnCl_2_, ZnCO_3_, Zn_3_(PO)_4_ and Ca_3_(PO)_4_. The corrosion mechanism and the formation of byproducts are schematically shown in Figure 3 [33].

The chemical composition and processing procedures (such as extrusion, cold rolling, hot rolling, etc.) of Zn alloys can play a major role in the corrosion mechanism and nature of the corrosion byproducts [29]. This is because the corrosion behavior of Zn alloys is affected by the size, distribution and volume fraction of both the Zn matrix and secondary phases that act as the cathodic sites during corrosion in SBF [34]. The microstructural characteristics of Zn alloys such as thermomechanical deformation can be altered during processing. Tong et al. [23] reported the corrosion and degradation behaviors of a series of Zn-1 Mg alloys that were alloyed with different rare-earth elements (REE), i.e., Zn–1 Mg–0.1 RE (wt.%) (RE = Er, Dy, and Ho), in both the as-cast and hot-rolled conditions for biomedical applications. These Zn alloys showed different corrosion and degradation properties due to the different microstructural characteristics and addition of different REEs. Figure 4 shows the corrosion behavior of as-cast and hot-rolled Zn–1 Mg and Zn–1 Mg–0.1 RE alloys tested in Hank’s solution. The PDP curves are shown in Figure 4a and the degradation rates after immersion for one month are shown in Figure 4b. Based on the PDP curves (Figure 4a), the corrosion rates calculated using the Tafel extrapolation method were ~378 μm/y, ~343 μm/y, ~363 μm/y and ~229 μm/y, respectively, for the as-cast Zn–1 Mg, Zn–1 Mg–0.1 Er, Zn–1 Mg–0.1 Dy and Zn–1 Mg–0.1 Ho, and ~247 μm/y, ~338 μm/y, ~328 μm/y and ~247 μm/y for the hot-rolled Zn–1 Mg, Zn–1 Mg–0.1 Er, Zn–1 Mg–0.1 Dy and Zn–1 Mg–0.1 Ho, respectively. Similarly, both the as-cast and hot-rolled Zn–1 Mg exhibited higher degradation rates than the Zn–1 Mg–0.1 RE alloy after immersion in Hank’s solution for one month, demonstrating REE addition to improve the corrosion resistance of Zn–1 Mg. In addition, the degradation rates of the hot-rolled Zn–1 Mg and Zn–1 Mg–0.1 RE alloys were faster than that of their as-cast counterparts, and the as-cast and hot-rolled Zn–1 Mg–0.1 Ho alloy exhibited the lowest degradation rate of ~0.038 mm/y and ~0.042 mm/y among the as-cast and hot-rolled Zn–1 Mg–0.1 RE alloys, while the as-cast and hot-rolled Zn–1 Mg showed the highest degradation rates of ~0.059 mm/y and ~0.074 mm/y, respectively [23]. Such variations are attributed to the differences in composition and processing procedures of the Zn alloys that caused different microstructural characteristics of the matrix and secondary phases.

The in vivo corrosion and degradation behaviors of Zn alloys can be assessed using various animal models such as Sprague–Dawley (SD) rats, Wistar rats, mice, beagle dogs, white pigs and rabbits [29]. Zn-alloy samples can be implanted in different sites in relation to their applications. For instance, Zn-0.1Li (wt.%) and pure Zn (>99.99% purity) wire samples with 0.25 mm diameter and 2 cm length were poked into the abdominal aorta and then directed within the lumen for 10 mm before exteriorization for up to 12 months to assess their suitability for cardiovascular stent applications [35]. Furthermore, as-cast, rolled and extruded pins of Zn-1X (wt.%) (X = Mg, Ca and Sr) of 0.7 mm in diameter and 5 mm in length were implanted in the mouse femora for up to 8 weeks for evaluation of orthopedic applications [26]. Moreover, Yang et al. [1] used a rat femur model to assess the in vivo performance of binary Zn-X (X = Mg, Ca, Sr, Li, Mn, Fe, Cu and Ag) alloys. The radiographs and reconstructed Micro-CT 3D images of the Zn-X implants post-surgery and at eight weeks after implantation are shown in Figure 5. The radiographs did not show gas shadows in the femoral condyle and bone marrow cavity adjacent to the implants at eight weeks and none of the implants showed obvious degradation at this timepoint.

There is little reported on the CF and SCC of Zn alloys under physiological conditions, which are critical for bioimplant performance (as evident from the literature [12,13,14]). In fact, a recent review on Zn alloys for biodegradable implant applications [1] emphasized such knowledge gap in the literature and the importance of generating related data.

## 5. Human Body-Fluid-Assisted Fracture of Implants: A Serious Concern

The simultaneous presence of dynamic loading along with corrosive environment can result in stress corrosion cracking (SCC) and corrosion fatigue (CF), which often occur at stresses considerably below design stresses that would be considered for a non-corrosive environment. The most fundamental and detrimental feature of CF and SCC is that a ductile material that would have undergone considerable elongation before fracture may suffer embrittlement in the presence of a corrosive environment, leading to premature brittle fracture. Because brittle SCC/CF fractures can be sudden, catastrophic and premature, they are the most dangerous forms of corrosion-assisted failures in bioimplants. The dynamic loading in the human body, along with the corrosive physiological environment accentuates the threat of CF and SCC. CF and SCC are of particular concern for devices with sharp contours, such as pins, screws and stents since sharp locations are common stress concentrators, and thus the crack initiation points. There have been several instances of fracture due to SCC of implants for traditional alloys (i.e., stainless steels, Ti alloys and Co–Cr alloys [12,13,14]). Such catastrophic failures would generally necessitate removal of fractured devices and painful irritation or inflammation of the surrounding tissue. CF and SCC are expected to be a serious concern for implant devices of Zn alloys, because: (a) Zn alloys can suffer pitting in aqueous chloride solutions [36] such as human body fluid, and pits are the most common initiators of CF and SCC; and (b) common temporary implant devices (such as screws, pins and plates) have sharp contours that can be a location for the initiation of CF and SCC. However, reported literature on human body-fluid-assisted fracture of Zn alloys is limited to just one recent publication [37].

### 5.1. Human Body-Fluid-Assisted Fracture of Zn Alloys

In the reported study on the assessment of SCC susceptibility of a zinc alloy in simulated body fluid (SBF) [37], a Zn-0.8Li alloy was subjected to slow strain-rate testing (SSRT) at a rate of 10^−5^ s^−1^ and constant load testing while immersed in circulating SBF at 37 °C. The study made a benign claim for the occurrence of SCC on the basis of a comparison of stress-strain data generated for the alloy in SBF and air, but did not make a conclusive inference on SCC. In general, it is critical to understand the role of strain rate in the susceptibility of an alloy to SCC. The strain rate employed in SSRT for investigating SCC is often in the range of 10^−6^–10^−7^ s^−1^, as established for various alloy systems such as Mg alloys in SBF [8,38,39,40] and steels in aqueous corrosive solutions [41,42]. Therefore, the strain rate of 10^−5^ s^−1^ that was employed for SSRT in the recent study on SCC of Zn-0.8Li [37] may be considered too high to cause SCC. It should be noted that when the strain rates are higher than that required for SCC, the alloy may still show a loss in mechanical strength or elongation when tested in a corrosive environment (compared to air), simply due to the corrosion-assisted loss of the cross-sectional area of the specimen, but such a loss in mechanical strength or elongation is not called SCC. In that case, the strain rate is too high and hence there is not enough opportunity/time for corrosion to occur at the crack tip for the required synergy of stress and corrosion that is essential for SCC. These aspects are clearly established for a Mg alloy subjected to SSRT in SBF and air in Ref. [40], where the alloy was tested at strain rates in the range of 1.2 × 10^−7^–3.1 × 10^−7^ s^−1^ and suffered SCC, as confirmed by representative fractographs revealing transgranular crack propagation and crack branching in SBF (Figure 6). As shown in Figure 7, such features were absent when SSRT tests were carried out at higher strain rates [40], i.e., although the alloy showed loss in mechanical strength and elongation, the fractographs confirmed the absence of SCC.

It is further relevant to add that SCC does also not take place for a given alloy-environment combination when the strain rates are too low, because at too low a strain rate the required synergy for stress and corrosion that is essential for SCC is again not achieved [42]. In this situation, corrosion takes over and the excessive corrosion blunts the crack tip (instead of letting it propagate). As shown in Figure 8a, a steel exposed to a high-temperature caustic solution clearly and reproducibly suffered SCC during SSRT when strain rates were in the vicinity of 3.1 × 10^−7^ s^−1^, but no SCC was observed when the strain rates were either higher or lower than 3.1 × 10^−7^ s^−1^. These findings have been corroborated by the fractographic features (Figure 8b,c).

In light of the profound role of strain rate in SCC, as established in the preceding discussion, it is important that the potential occurrence of SCC in the recent study [37] is verified by carrying out SSRT investigations at various strain rates before a conclusive inference may be made on the susceptibility of zinc alloys to SCC.

### 5.2. Circumventing SCC and CF Fracture of Zn Alloys: Challenges and Opportunities

Alloys of highly anodic metals (such as Zn and Mg) often suffer extensive pitting due to secondary precipitates that are commonly developed in alloys for the purpose of strengthening [22]. Most secondary precipitate types in such alloys are highly cathodic to the alloy matrix causing a micro-galvanic effect and localized corrosion/pitting in the adjacent matrix, as shown in Figure 2. As emphasized earlier, pits are the most common sites for initiation and propagation of environment-assisted fracture, i.e., CF and SCC. So, an effective strategy for circumvention of human body-fluid-assisted fracture (i.e., CF and SCC) of Zn alloys necessitates removal/minimization of secondary precipitates. This can be achieved by keeping the alloying contents below their solubility limits. However, this task may be difficult for Zn alloys since the solubility of most non-toxic/bio-safe alloying elements in Zn is negligible [43] (except for a few alloying elements such as Mn and Ag). Furthermore, Zn is susceptible to ageing and recrystallization even at ambient temperatures [44,45] as well as to strain-induced precipitation [46].

In light of the restrictions described above, suppression of precipitation in Zn alloys may be a challenging task. However, there is great value in developing Zn-alloy implants that possess adequate resistance to human body-fluid-assisted fractures. Therefore, there may be a need to:(a)Maximize solid-solution strengthening by selecting alloying elements that have high solubility and also meet the non-toxicity criteria for body implant applications;(b)Employ suitable thermomechanical treatments to control the grain size as a complementary strengthening measure.

For example, among the alloying elements that meet the non-toxicity criterion, manganese (Mn) possesses significant solubility in Zn [22], and hence may be among the choices as principal alloying element. In fact, Mn is an essential enzyme activator for the human body and its deficiency can cause osteoporosis, diabetes mellitus, and artherosclerosis; the Mn content in the human body is 12 mg [46,47]. Zn–Mn alloys may require the addition of other minor alloying elements in order to control the microstructure. The minimization of secondary precipitation for the purpose of avoiding pitting and still achieving the required strength has been successfully demonstrated for extruded “lean” MgZnCa alloys (ZX alloys) [48,49,50]. In that type of alloys, it was also possible to tailor the electrochemical character of the secondary precipitates from cathodic (Ca_2_Mg_5_Zn_5_ [50]) to anodic (Mg_2_Ca), for manipulating pitting susceptibility. A similar strategy may be pursued for Zn alloys. However, it is noted that the strain-hardening characteristics of Zn alloys during extrusion will be considerably different, and the processing parameters will thus need to be thoroughly optimized.

## 6. Appropriate Mechano-Chemical Conditions for Corrosion-Assisted Fracture

A proper simulation of the chemical composition of the body fluid and ensuring that mechanical factors have been appropriately considered is critical for generating accurate data for corrosion and corrosion-assisted fracture of bioimplants.

### 6.1. Pseudo-Physiological Test Solutions Simulating Organic Constituents

As elaborated earlier, in vitro corrosion studies on biodegradable metals in simulated body fluids (SBFs) have employed many types of pseudo-physiological solutions that mimic the composition of body fluids [51,52,53]; for example, Hank’s solution, modified simulated body fluid (m-SBF), and Dulbecco’s modified Eagle medium (DMEM). Most of these studies used SBFs that simulated only the inorganic constituents of the body fluid, whereas physiological environments also contain organic compounds such as proteins, amino acids and glucose. In one of the few studies that also simulated the organic constituents of body fluid, Witte et al. demonstrated that “for in vitro corrosion tests ASTM standards cannot be used to predict corrosion rates of Mg alloys in vivo” [6]. Yang et al. [54] reported “huge variations in corrosion rates depending on the corrosion environment; proteins delayed corrosion and altered the ion composition of the solutions, and the choice of appropriate corrosion environment is crucial for in vitro experiments”. Yamamoto et al. [55] showed that protein adsorption in the barrier layer retarded Mg degradation, whereas amino acids accelerated it; and Xin et al. [56] also reported the adsorption of proteins to improve corrosion resistance. Therefore, organic constituents of body fluid are among the critical factors in performing appropriate in vitro studies on biodegradable alloys. Albumin, globulin and fibrinogen are the main proteins present in blood plasma [54], and serum albumins constitute up to ~55% of blood proteins. The normal concentration of albumins in blood is 30–50 g/L. Bovine serum albumin (BSA) and human serum albumin (HSA), which are derived from bovine and human blood respectively, are frequently used in biophysical and biochemical studies.

Another critical aspect concerns the appropriate means for controlling the pH of the simulated body fluid (SBF), which is commonly achieved using buffering chemicals, such as 2-(4-(2-hydroxyethyl)-1-piperazinyl) ethanesulfonic acid (HEPES)). However, HEPES can interfere with the main corrosion reaction [57]. The actual human body fluid controls its pH by regulating the supply of CO_2_. As shown in Figure 9, a device for minutely controlled CO_2_ bubbling through SBF (instead of HEPES) to control pH was employed in recent studies on CF (and SCC) of biodegradable alloys [57,58].

Metallurgical characteristics such as grain size and texture, which develop as a result of thermomechanical processing such as extrusion, have been found to influence the corrosion of magnesium alloys [59], and hence these variations can also influence SCC of such alloys [10], in particular in physiological environments. However, such aspects have not been investigated for zinc alloys.

Little has been reported on the influence of physiological molecules such as proteins and glucose content of blood plasma in corrosion and corrosion-assisted cracking (i.e., SCC and CF) of Zn alloys. In particular, there are no reported studies on SCC and CF of such alloys in SBF that considered both inorganic and organic constituents of body fluid. A study that investigated SCC and CF of a Mg alloy in SBF with and without the most abundant organic constituent of body fluid, i.e., bovine serum albumin (BSA), found BSA addition to significantly increase susceptibilities to SCC but, intriguingly, BSA addition retarded corrosion fatigue [58]. Such studies are rare even for Mg alloys, and apparently non-existent for Zn alloys. Thus, it is important to establish the right environmental conditions for human body-fluid-assisted fracture tests on Zn alloys that will be identified via an optimized composition and strengthening process (as described earlier).

### 6.2. Appropriate Mechanical Tests to Achieve Relevant SCC/CF Data for Accurate Design of Bioimplants

Another aspect that needs particular attention is that the devices often consist of both sharp and smooth contours. While SCC and CF can also initiate on smooth metallic surfaces, sharp contours are the most susceptible locations for SCC and CF initiation. Therefore, the most common SCC and CF data used for design purposes are generated using specimens with pre-existing sharp cracks/notches. Smooth specimens are tested for establishing the susceptibility of a metallic material to cracking, whereas tests using specimens with pre-existing sharp cracks/notches establish the susceptibility of an existing crack to propagate under the influence of a corrosive environment. Therefore, it is essential to test pre-cracked specimens of identified Zn alloys, and such data are currently non-existent for Zn alloys in body fluid. Circumferential notch tensile (CNT) testing [60] is a cost-effective approach for determining the susceptibility/threshold of an existing crack/notch/sharp contour to propagate in SCC mode (i.e., *K*_ISCC_). The scientific/mechanistic basis for the CNT technique to generate accurate *K*_ISCC_ data has also been validated. The CNT testing and associated specimen fabrication is simpler and less expensive (~25% of the cost of traditional testing) [61], which thus allows for the large number of tests that are often required to generate robust design data.

## 7. Conclusions

Because corrosive body-fluid-assisted premature fracture is a concern for bioimplants, it is necessary to understand and address such fracture before implants of such materials are put into actual service. This article briefly discusses the suitability of Zn, Mg, and Fe alloys for their use as biodegradable implant materials, and critically assesses recent investigations on stress corrosion cracking (SCC) of Zn alloys as potential bioabsorbable implant materials. One of the critical aspects in this article is to illustrate the need of comprehensive investigations of SCC and corrosion fatigue (CF) of Zn alloys in physiological environments. This article identifies in particular the critical knowledge gaps in SCC and CF investigations on Zn alloys in appropriately simulated mechanical loading and human body-fluid conditions.

## Figures and Tables

**Figure 1 materials-16-04984-f001:**
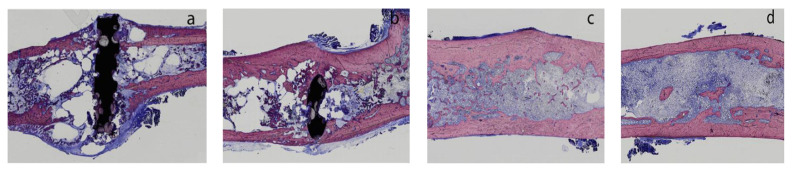
Pins of a magnesium alloy ZX50 implanted into femoral bone tissues of Sprague–Dawley rats, after different durations (i.e., (**a**–**d**): 4, 12, 24, 36 weeks), showing complete dissolution of the alloy and joining of a fractured bone within 36 weeks. Reprinted with permission from [5], 2012, Elsevier.

**Figure 2 materials-16-04984-f002:**
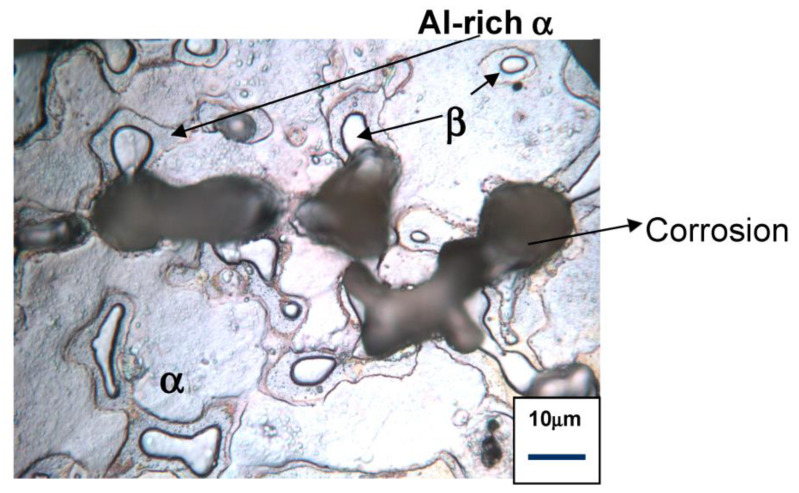
Secondary precipitates (β) causing pitting corrosion of the adjacent Mg alloy matrix (α) during exposure to aqueous chloride solution [9].

**Figure 3 materials-16-04984-f003:**
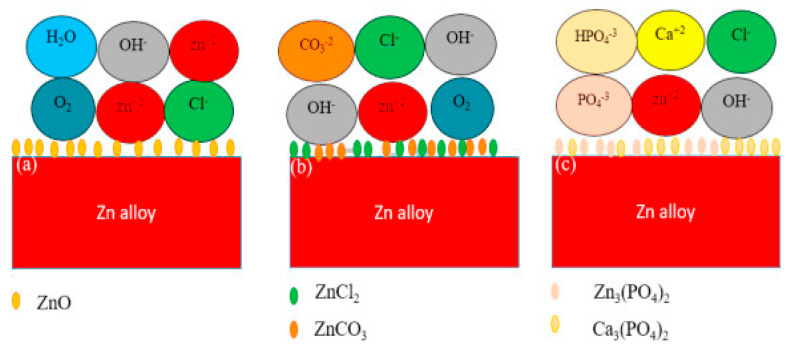
Schematic of corrosion mechanism of Zn immersed in SBF: (**a**) formation of ZnO; (**b**) formation of ZnCl_2_ and ZnCO_3_; and (**c**) formation of Zn_3_(PO)_4_ and Ca_3_(PO)_4_. Reproduced with permission from Ref. [33].

**Figure 4 materials-16-04984-f004:**
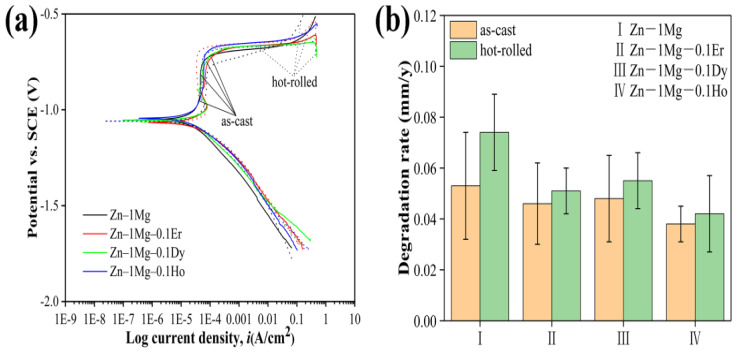
Corrosion behaviors of as-cast and hot-rolled Zn–1 Mg and Zn–1 Mg–0.1 RE alloys tested in Hank’s solution: (**a**) PDP curves and (**b**) degradation rates after immersion for one month. Reproduced with permission from Ref. [23].

**Figure 5 materials-16-04984-f005:**
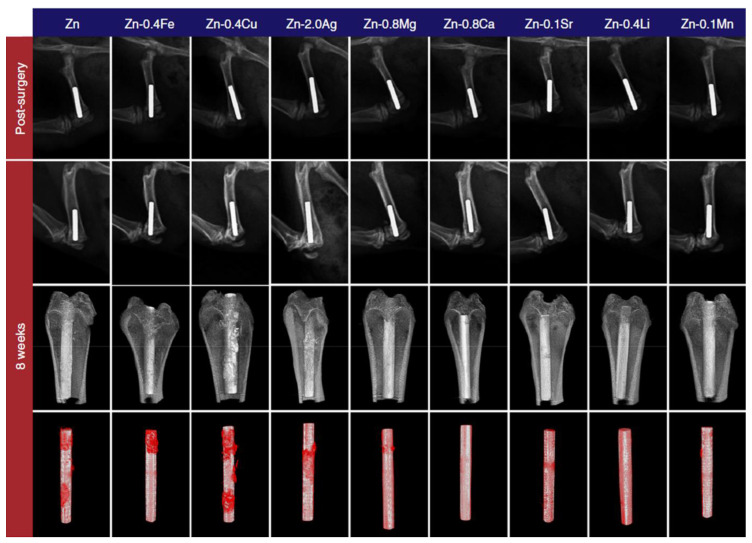
Radiographs of Zn-X (X = Mg, Ca, Sr, Li, Mn, Fe, Cu, and Ag) alloy implants in rat femurs. First row: post-surgery; second row: eight weeks after implantation; third row: 3D reconstructions with bone tissue at eight weeks; and fourth row: 3D reconstructions without bone tissue at eight weeks. Degradation products are marked in red. Reproduced with permission from Ref. [1].

**Figure 6 materials-16-04984-f006:**
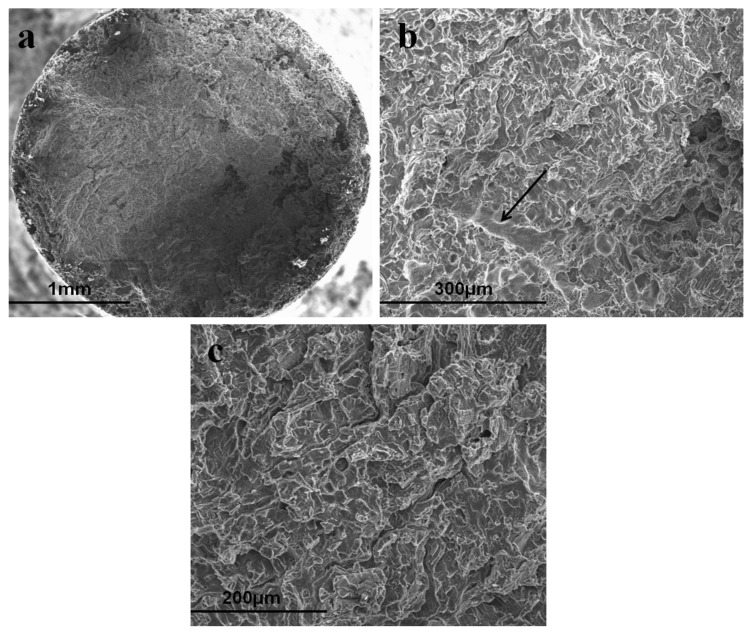
Fractographs of an AZ91 alloy subjected to slow strain-rate testing (SSRT) in SBF at a strain rate of 1.2 × 10^−7^ s^−1^. (**a**) Overall fracture surface with evidence of pitting at the specimen circumference; (**b**) transgranular SCC (indicated by arrow) on a considerable fraction of the fracture surface; and (**c**) crack branching [40].

**Figure 7 materials-16-04984-f007:**
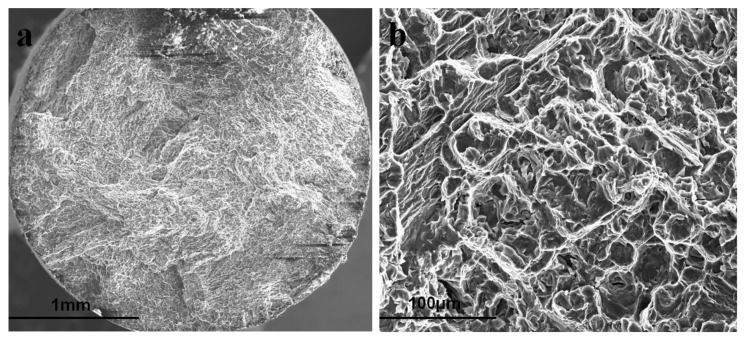
Fractographs of an AZ91 alloy subjected to slow strain-rate testing (SSRT) in SBF at a strain rate of 5.0 × 10^−7^ s^−1^. (**a**) Overall fracture surface, and (**b**) only ductile dimples (and no features for SCC) over the entire fracture surface [40].

**Figure 8 materials-16-04984-f008:**
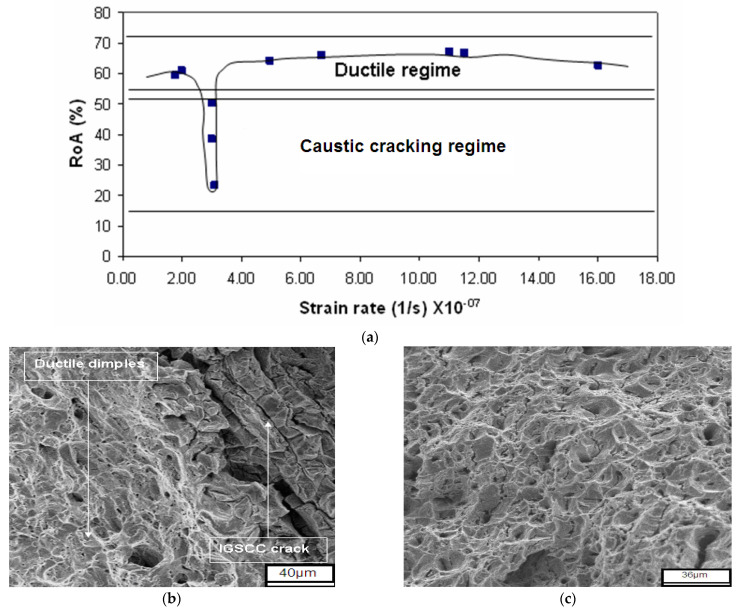
Slow strain-rate testing (SSRT) of mild steel in 20% NaOH solution at 120 °C: (**a**) Plot of reduction of area (RoA) at various strain rates; (**b**) SEM fractograph showing intergranular SCC (IGSCC) of the specimen tested at a strain rate of 3.1 × 10^−7^ s^−1^; and (**c**) representative SEM fractograph showing only ductile dimples (and no IGSCC) on the entire fracture surface of the specimen tested at strain rates higher or lower than 3.1 × 10^−7^ s^−1^ [42].

**Figure 9 materials-16-04984-f009:**
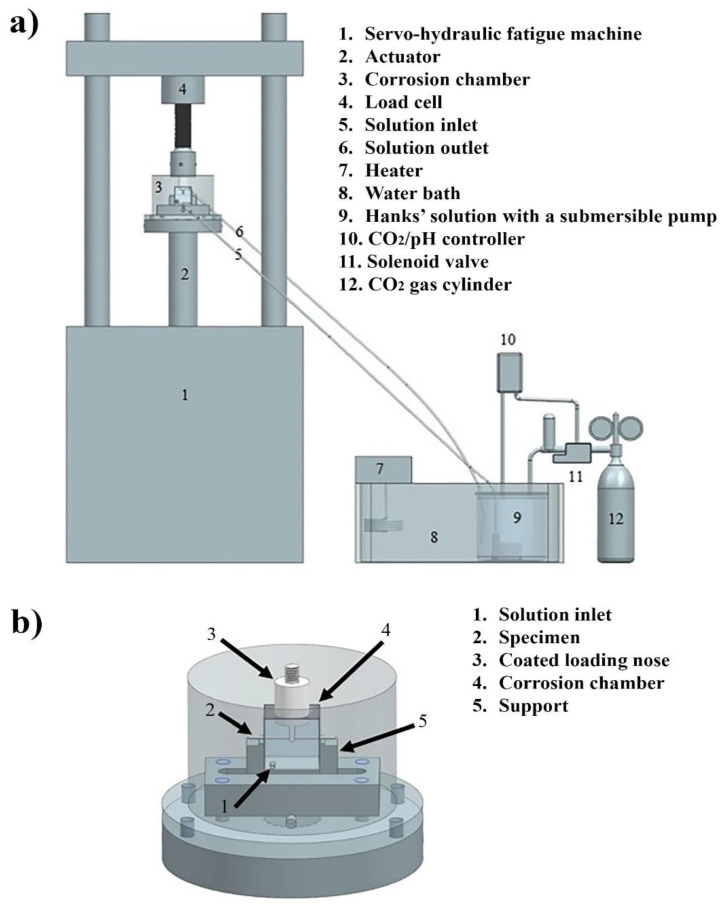
(**a**) Schematic of experimental set-up for a three-point cyclic bending test, with arrangement for pH control of the test solution through CO_2_ supply. (**b**) Specific fixture in the three-point bending corrosion fatigue chamber [58].

## Data Availability

Not applicable.

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
