# Peer review of "Human Body-Fluid-Assisted Fracture of Zinc Alloys as Biodegradable Temporary Implants: Challenges, Research Needs and Way Forward"

_materials, 2023, doi:10.3390/ma16144984_

Round 1

Reviewer 1 Report

This paper is an overview of the challenges, research needs and way forward in understanding human body fluid-assisted fracture of Zn alloys in human body fluid. However, it has a large discrepancy with the requirements of the journal, and is not recommended to be accepted.

1.       The title of this paper is human body fluid-assisted fracture of Zinc alloys as biodegradable temporary implants: challenges, research needs and way forward. The subject should focus on fracture of zinc alloys caused by corrosion, but the paper keeps introducing material properties such as degradation of zinc alloys and fracture caused by corrosion. The discussion is not focused and not in-depth.

2.       The corrosion behavior of the material has a strong influence on the fracture of zinc alloys induced by simulated body fluids, and how the authors consider the different corrosion behaviors of zinc alloys such as pitting and filiform corrosion?

3.       Besides the corrosive environment, alloying elements and grain size also has an influence on the fracture, how did the author consider about this?

4.       The abstract section is too simple and needs to be rewritten.

Discussion and abstract sections need to be strengthened

Author Response

Please refer to the attached doc.

Reviewer 2 Report

The review article entitled "Human Body Fluid-assisted Fracture of Zinc Alloys as Biodegradable Temporary Implants: Challenges, Research Needs and Way Forward" discusses the suitability of Zn-, Mg- and Fe-alloys for their use as biodegradable implant materials, and critically assesses the recent investigations on stress corrosion cracking (SCC) of Zn alloys as potential bioabsorbable implant materials. However, the authors failed to go in depth and have their own input in discussing relevant articles.

The authors need to go more deep and focus on the most successful/applicable or feasible inventions.

The abstract is too short and need to be rewritten and elaborated.

Authors have put many references in the manuscript but they are not implemented correctly. They also need to include more recent references and I suggest these two references to be included: https://doi.org/10.3390/biomimetics7040242, https://doi.org/10.3390/molecules28010306

The authors prospective and opinions need to be included in the conclusion section.

The quality of figures 2, 3 and 4 need to be improved.

The objective of this review needs to be added and included in the introduction section.

Moderate editing of English language Is required.

Author Response

Please refer to the attached doc.

Reviewer 3 Report

Dear Authors, your manuscript is indeed interesting and topical. It would be prudent to pay attention to a few issues during the revision. 

-Although the manuscript is well written and easy to understand, there are still numerous grammatical mistakes. It is recommended that the manuscript be edited by a native English Language speaker or professional editing service provider.

-In several instances through out the manusript the authors did not specify the relevant metal species they were refering to. For example: Line 108 "Fe is non-toxic, biocompatible and biodegradable in the human body." This statement is neither true nor accurate without providing relevant details as well as the corresponding reference. Issues like this should be corrected throughout the manuscript.

-The images are of poor quality and should be replaced with images of better resolution.

Best wishes!

The manuscript is replete with grammatical errors and the quality of English language needs improvement.

Author Response

Please refer to the attached doc.

Reviewer 4 Report

I recommend the manuscript can be accepted for publication.

Minor English revision is required before publication.

Reviewer 5 Report

Dear Authors,

Please find below my comments/observations regarding your manuscript:

1. Please add an explanation for figure 2, and for the figure numbered in the text with the figure 2, which should become 3, please add a magnification.

2. p.4, line 107: “(a) they corrode too rapidly”. Please add an explanation about what you mean - too rapidly: faster than bone healing.

Round 2

Reviewer 1 Report

Many papers investigate the influence of texture, twins and grain size on the corrosion resistance of zinc alloys. In contrast, the author considers that " However, such aspects have not been investigated for zinc alloys."  The paper needs to be deep enough in discussion section.

Many papers investigate the influence of texture, twins and grain size on the corrosion resistance of zinc alloys. In contrast, the author considers that " However, such aspects have not been investigated for zinc alloys."  The paper needs to be deep enough in discussion section.

Reviewer 2 Report

The authors have failed to address the comments properly.

Moderate editing of English language.
